# Active Rumen Bacterial and Protozoal Communities Revealed by RNA-Based Amplicon Sequencing on Dairy Cows Fed Different Diets at Three Physiological Stages

**DOI:** 10.3390/microorganisms9040754

**Published:** 2021-04-02

**Authors:** Lucia Bailoni, Lisa Carraro, Marco Cardin, Barbara Cardazzo

**Affiliations:** Department of Comparative Biomedicine and Food Science (BCA), University of Padova, Viale dell’Universitá 16, 35020 Legnaro, PD, Italy; lisa.carraro@unipd.it (L.C.); marco.cardin.4@phd.unipd.it (M.C.); barbara.cardazzo@unipd.it (B.C.)

**Keywords:** physiological stages, protozoa, RNA, cDNA, active rumen microbiota

## Abstract

Seven Italian Simmental cows were monitored during three different physiological stages, namely late lactation (LL), dry period (DP), and postpartum (PP), to evaluate modifications in their metabolically-active rumen bacterial and protozoal communities using the RNA-based amplicon sequencing method. The bacterial community was dominated by seven phyla: Proteobacteria, Bacteroidetes, Firmicutes, Spirochaetes, Fibrobacteres, Verrucomicrobia, and Tenericutes. The relative abundance of the phylum Proteobacteria decreased from 47.60 to 28.15% from LL to DP and then increased to 33.24% in PP. An opposite pattern in LL, DP, and PP stages was observed for phyla Verrucomicrobia (from 0.96 to 4.30 to 1.69%), Elusimicrobia (from 0.32 to 2.84 to 0.25%), and SR1 (from 0.50 to 2.08 to 0.79%). The relative abundance of families Succinivibrionaceae and Prevotellaceae decreased in the DP, while Ruminococcaceae increased. Bacterial genera *Prevotella* and *Treponema* were least abundant in the DP as compared to LL and PP, while *Ruminobacter* and *Succinimonas* were most abundant in the DP. The rumen eukaryotic community was dominated by protozoal phylum Ciliophora, which showed a significant decrease in relative abundance from 97.6 to 93.9 to 92.6 in LL, DP, and PP, respectively. In conclusion, the physiological stage-dependent dietary changes resulted in a clear shift in metabolically-active rumen microbial communities.

## 1. Introduction

The bovine rumen is colonized by a complex microbial ecosystem comprising symbiotic populations of bacteria, ciliated protozoa, fungi, and archaea [1]. These symbionts enable the ruminants to convert indigestible fibrous plant mass into nutrient sources such as volatile fatty acids, lipids, amino acids, lactate, and hydrogen that are essential for the growth, production performance, and health of the ruminants [2]. The composition of rumen microbiota is affected by several factors, such as diet [3,4,5], age and health [6,7], breed [8], diet and age [9], environment, and host genetics [10]. Diet composition is found to have a major influence on the community structure and fermentation patterns of rumen microbes [11], which in turn affect methane production, host health, and productivity [12].

Modification of the diet is required for gestating cows during the dry period (prepartum). The effective feeding management strategy for better transition success is to dilute the high-energy forage diet with low-energy forage components such as wheat straw [13] or supplementation of high-forage diet with fat [14]. The strategy of using energy-diluted diets results in higher neutral detergent fiber (NDF) and lower energy content of the diet, which may reduce the host energy and total dry matter intake, maintain growth rate, and can prevent postpartum cows from liver fat deposition [15,16].

Such diet modifications are expected to have a great impact on the ruminal microbiota that is highly associated with host feed digestion and metabolism [17]. Nevertheless, few comparative studies on the effect of physiological stage-dependent dietary modifications on community profile of rumen microbiota are available, and this area must be explored to define better feeding management strategies [18,19,20,21]. In recent years, culture-independent analyses, such as metabarcoding, have been largely applied and have greatly expanded our understanding of the rumen microbial communities [22]. These approaches have allowed for the complete definition of the microbial community, including non-cultivable species that have been estimated to represent 90% of the rumen microbiota [23]. The majority of the published rumen microbiome data is based on sequencing of 16S rRNA genes amplified from a DNA template [24,25,26]. However, DNA-based methods are not reliable when microbial activity is concerned, as these methods are unable to distinguish between the genes that come from active, inactive, lysed, or dead cells [27], thus making it impossible to predict the viability and biological and metabolic activity of the detected microbial communities [28]. On the contrary, RNA-based methods are considered accurate predictors of metabolically-active microbial community structures [29,30] due to the presence of a positive correlation between the content of ribosomal RNA (rRNA) and metabolic activity of microorganisms [31]. Recent comparisons between DNA- and RNA-based rumen microbial community profiles under different dietary treatments resulted in the identification of certain distinct phylotypes and unique microbial taxa for each approach [32,33,34]. RNA-based analysis showed a higher dominance of the phylum Proteobacteria in grain-fed cows than in DNA-based datasets [32]. Similarly, a high number of *Prevotella* transcripts were observed at the RNA-level using a higher proportion of corn silage in the dairy cow diet; however, no effects on rumen microbial communities were observed at the DNA-level [33]. Hence, it is worth noting that RNA-based studies may provide a different community profile from DNA-based studies when evaluating potentially active rumen microbiota. The present study was conducted to evaluate the effect of diet-dependent modifications in active rumen bacterial and protozoal community structure using an RNA-based method. This was done by total RNA-extraction, cDNA synthesis, and subsequent amplification of RNA-derived transcripts–amplicons by PCR using primers specific for bacterial 16S rRNA and eukaryotic 18S rRNA genes, followed by sequencing using the Illumina Miseq platform. This approach allowed for a comparison of the relative amount of the active microbial taxa in the different physiological stages. To the best of our knowledge, this study is the first of its type to report modifications in the metabolically active rumen bacterial and protozoal community structure covering the overall physiological stages of dairy cows.

## 2. Materials and Methods

### 2.1. Ethics Statement

All experimental procedures were carried out according to Italian law on animal care (Legislative Decree No. 26 of 14 March 2014) and approved by the ethics committee at the University of Padova (approval number 6/2020).

### 2.2. Animal Management, Diet, and Rumen Fluid Sampling

Seven lactating Italian Simmental cows (3 primiparous, 1 secondiparous, and 3 multiparous) were used as rumen fluid donors in this study and were housed under the same rearing environment at the Experimental Farm of University of Padova, Italy. At the beginning of the experiment, all cows were pregnant (from 148 to 203 days in gestation). The experiment was composed of three phases based on physiological stages of dairy cows: LL (248–332 days in milk), DP (8–46 days pre-calving), and PP (15–38 days post-calving). Cows were fed a specific total mixed ration (TMR) formulated to cover their nutritional requirements (Table 1). This was done for at least two weeks before sample collection, during each physiological stage. Rumen fluid samples were collected between 6 am and 7 am, before morning feeding, using an esophageal vacuum pump system as described by Tagliapietra et al. [35]. One sample (approximately 500 mL) of rumen fluid (mixed with fine feed particles) was collected from each cow for each physiological stage. The samples were immediately filtered through four layers of cheesecloth to remove feed particles and transferred to the laboratory under controlled temperature conditions using preheated (39 ± 0.5 °C) thermal flasks. Samples were homogenized, and an aliquot of 50 mL was directly used in further analysis.

### 2.3. RNA-Extraction and Synthesis of cDNA

A total volume of 50 mL of freshly filtered rumen fluid samples were pelleted by centrifugation at 2683× *g* for 10 min. Supernatants were discarded, and pellets were re-suspended in phosphate-buffered saline (PBS) to obtain a final concentration of 1 g rumen content/1 mL of the re-suspended mixture. Re-suspended mixture (100 μL) was mixed with TRIZOL reagent (1 mL), and cells were disrupted using zirconia silica beads (0.25 mm) on a TissueLyser (Qiagen, Hilden, Germany). Disruption was carried out in high-speed mode 4 times for 30 s at 30 Hz in two rounds of bead beating steps (30 frequency 1/s, 30 s each). Chloroform (200 μL) was added and each sample was vortexed for 15 s, left on ice for 15 min, and pelleted by centrifugation at 12,000× *g*, 4 °C for 15 min. The extracted RNA was purified using an RNeasy Mini Kit (Qiagen), including an on-column DNase digestion step according to the manufacturer’s protocol. Following centrifugation, the upper aqueous phase (350 μL) was transferred to a fresh tube, and 100% ethanol (350 μL) was added. This mixture (700 μL) was transferred to the column provided in the RNeasy Mini Kit (Qiagen) and centrifugated at 18,407× *g* for 30 s. RNA concentration was assessed using a Nano kit (Agilent Technologies, Santa Clara, CA, USA). Purified RNA was reverse-transcribed into single-stranded cDNA using the SuperScript IV cDNA synthesis protocol (ThermoFisher Scientific).

### 2.4. Targeted Amplicon Sequencing of Bacteria and Protozoa

Before PCR amplification, the cDNA templates (containing 1 μg cDNA/µL) were diluted with Rnase-free water to obtain a final concentration of 40 ng cDNA/µL. In order to make the amplicons compatible with Illumina MiSeq sequencing, overhang adapter sequences were additionally linked to the primers. For analysis of bacterial communities, tailed primers specific for V3–V4 hypervariable region of the 16S rRNA gene were used, forward primer = Pro341F: 5′-TCG TCG GCA GCG TCA GAT GTG TAT AAG AGA CAG CCT ACG GGN BGC ASC AG-3′; reverse primer = Pro805R: 5′-GTC TCG TGG GCT CGG AGA TGT GTA TAA GAG ACA GGA CTA CNV GGG TAT CTA ATC C-3′ [36]. The PCR mixture was composed of 5× high fidelity PCR buffer (5 µL), 25 mM dNTPs (0.2 µL), 10 µM forward primer (1 µL), 10 µM reverse primer (1 µL), 2 U/μL Phusion high-fidelity DNA polymerase (0.5 µL; Thermo-Fisher Scientific, Waltham, MA, USA), 40 ng cDNA template (5 µL) to a final volume of 25 µL with Rnase-free water. Thermocycling conditions for PCR comprised an initial denaturation at 94 °C for 1 min, followed by 25 cycles of annealing at 95 °C for 30 s, 55 °C for 30 s, and 68 °C for 45 s and a final extension at 68 °C for 7 min. PCR products were controlled on a 1.8% agarose gel.

The protozoal communities were analyzed using modified primers specific for V9 region of 18S rRNA gene from the Earth microbiome project [37,38], Illumina_Euk_1391f forward primer = TCGTCGGCAGCGTCAGA TGTGTATAAGAGACAGGTACACACCGCCCGTC, and Illumina_EukBr reverse primer = GTCTCGTGGGCTCGGAGATGTGTATAAGAGACAGTGATCCTTCTGCAGGTTCACCTAC. The PCR mixture was composed of 5× high fidelity PCR buffer (5 µL), 25 mM dNTPs (0.2 µL), 10 µM forward primer (0.5 µL), 10 µM reverse primer (0.5 µL), 2 U/μL Phusion high-fidelity DNA polymerase (0.5 µL; Thermo-Fisher Scientific, Waltham, MA, USA), 40 ng cDNA template (5 µL) to a final volume of 25 µL with Rnase-free water. Thermocycling conditions for PCR comprised initial denaturation at 94 °C for 3 min followed by 35 cycles of annealing at 94 °C for 45 s, 57 °C for 60 s, and 72 °C for 90 s, and final extension at 72 °C for 10 min. PCR products quality was controlled on a 1.8% agarose gel. PCR product indexing and sequencing were done by BMR Genomics, Padova (Italy) using the Illumina MiSeq platform with a paired-end 300-cycle run.

### 2.5. Bioinformatic Analysis

The bioinformatic analysis of the targeted amplicon sequencing datasets, covering the 16S and 18S rRNA gene, was very similar with few exceptions. Briefly, the Illumina MiSeq sequences (2 × 300 bp) were demultiplexed using CASAVA v1.8 (Illumina) software. In total 1,960,561 raw reads (16S rRNA amplicon sequence data) and 3,560,235 raw reads (18S rRNA amplicon sequence data) were generated, with 93,360 ± 5448 reads (mean ± SEM; 16S rRNA amplicon sequence data) and 169,535 ± 10,225 reads (mean ± SEM; 18S rRNA amplicon sequence data) per sample. The quality of the sequencing data was checked with FastQC program (version 0.11.7; https://www.bioinformatics.babraham.ac.uk/projects/fastqc/; accessed on 18 March 2021). The bioinformatic analysis was performed using QIIME 2 (2018.11) software [39]. The demultiplexed raw sequence data files were imported into QIIME 2 artifact using “SampleData [PairedEndSequencesWithQuality]” semantic type. In the case of 18S rRNA amplicon sequencing data, the residual artificial sequences (forward (16 bp) and reverse (24 bp) primers) were removed by implementing cutadapt v1.18 within the QIIME 2 artifact using q2-cutadapt plugin and trim-paired command [40]. Raw paired-end sequence data files were first trimmed using cutadapt linked adapters command-line, followed by --p-anywhere option. However, the removal of residual artificial sequences (unclipped forward (17 bp) and reverse (21 bp) primes) from 16S rRNA amplicon sequencing data was done during the DADA2 step. The trimmed sequence data files (belonging to 18S rRNA transcript amplicons), as well as untrimmed 16S rRNA amplicon sequence data files, were quality filtered (cut bases with an average quality score below 20). Denoised paired-end sequences were merged (mean length: 414 bp and 111 bp; 16S rRNA and 18S rRNA transcript amplicons, respectively). Non-overlapping regions, chimeric sequences, and singletons were discarded, and FeatureTable[Frequency] and FeatureData[Sequence] QIIME 2 artifacts were generated using q2-dada2 plugin which implements DADA2 pipeline within the QIIME 2 [41]. Taxonomic classification of 16S rRNA amplicon sequence data was performed using q2-feature-classifier plugin and a Naive Bayes classifier (pre-trained on SILVA 16S rRNA reference database (release_132) clustered at 99% similarity) [42]. Taxonomic classification was done using the Naive Bayes sklearn-based taxonomy classifier with a default confidence of 0.7. Non-bacterial, cyanobacteria and chloroplast, and unassigned sequences from FeatureTable[Frequency] and FeatureData[Sequence] QIIME 2 artifacts were removed with taxonomy-based filtering step using q2-taxa plugin in QIIME 2. A total of 641,525 reads were generated representing 4974 unique bacterial OTUs. All 4974 selected unique bacterial OTUs were taxonomically reassigned based on the Ribosomal Database Project (RDP) classifier (http://rdp.cme.msu.edu/; accessed on 18 March 2021) [43]. The RDP classifier taxonomy output table was filtered by setting up cutoff values of confidence thresholds to 94.5% for genus, 86.5% for family, 82.0% for order, 78.5% for class, and 75.0% for phylum [44]. The sequences belonging to the unclassified taxon within the dominant Proteobacteria phylum were rechecked manually for better taxonomic designations using RDP SeqMatch [45]. For 18S rRNA amplicon sequencing data, the taxonomic classification was performed using q2-feature-classifier plugin and classify-consensus-blast (an alignment-based classification method). This method used SILVA 18S rRNA (release_132) as a reference database with a confidence level of 0.90. Due to the lack of proper information about the seven taxonomic ranks in SILVA 18S rRNA reference database, the identified genera were rechecked using UniProt Knowledgebase (UniProtKB) (https://www.uniprot.org/taxonomy/; accessed on 18 March 2021) and Systema Naturae 2000 (http://sn2000.taxonomy.nl/; accessed on 18 March 2021) databases to obtain exact names of each taxonomic rank. Non-eukaryotic and unassigned sequences from FeatureTable[Frequency] and FeatureData[Sequence] were finally removed with the taxonomy-based filtering step using q2-taxa plugin in QIIME 2. A total of 2,194,288 reads were generated representing 256 unique eukaryotic OTUs.

The raw sequence data were deposited in the SRA database with accession number PRJNA680705.

### 2.6. Statistical Analysis

Statistical analyses were performed using Calypso version 8.84 [46] and Primer-e (PRIMER 6.1.16 and PERMANOVA+ 1.0.6) [47] software. For Calypso software, two QIIME 2 output files were exported, namely FeatureTable[Frequency] as feature-table.biom and FeatureData[Taxonomy], while for Primer-e software, a BIOM table (feature-table-with-taxonomy annotations) was created using the biom add-metadata command in QIIME 2. The diversity indices (richness, evenness, and alpha-diversity estimators) among amplicon sequence datasets were made comparable by rarefying samples to the lowest number of reads observed in each dataset (a read depth of 18,929; 16S rRNA amplicon sequencing dataset, and a read depth of 41,480; 18S rRNA amplicon sequencing dataset) (Table 2). The statistical differences among diversity indices of physiological stages were tested with a one-way analysis of variance (ANOVA) test, and for pair-wise comparisons, the Wilcoxon rank test was used in Calypso v8.84. The physiological stage-dependent modifications in the rumen active microbial community structure were confirmed using principal-coordinate analysis (PCO) and analysis of similarities (ANOSIM) based on Bray–Curtis dissimilarity matrices in Primer-e. The significantly different bacterial and protozoal taxa among physiological stages were identified using the one-way analysis of variance (ANOVA) test, and for pair-wise comparisons, Tukey’s test was used on normalized OTU abundance data (Appendix A). For the 16S rRNA amplicon sequencing dataset, OTU abundance data were normalized with the total sum normalization (TSS) method. However, for the 18S rRNA amplicon sequence dataset, the OTU abundance data was normalized using Hellinger transformation (total sum normalization (TSS) combined with square root transformation) prior to the calculation of Bray–Curtis dissimilarity matrices.

To describe bacterial and eukaryotic rumen communities, genus-level heatmaps were plotted using the heatmap.3 package in R (version 1.3.1093) on normalized data.

## 3. Results 

### 3.1. Effect of Physiological Stages of Dairy Cows on the Diversity of Their Active Rumen Microbiota

Physiological stage-dependent dietary amendments were found to have a clear effect on the diversity of active rumen bacterial and eukaryotic communities of Italian Simmental cows. The bacterial species richness (Chao 1) as well as evenness increased significantly during transition from LL to DP and also showed a non-significant (*p* > 0.05) minor decrease during passage from a DP to PP diet (Table 2). Similar to the bacteria, high-forage feeding in the DP also significantly increased the eukaryotic species richness (Chao 1) from a value of 47.0 in LL to 68.0 in the DP, followed by a significant decrease to 40.0 during transition to the PP diet. However, the eukaryotic species evenness remained unaffected (*p* > 0.05) by physiological stage-dependent dietary changes (Table 2).

Likewise, both alpha-diversity indices (Shannon and Inverse Simpson) showed a significant increase in the bacterial taxa diversity from the lowest diversity in LL (4.35 and 16.4) to the highest diversity in DP (5.50 and 86.6), followed by a significant decrease in diversity to 4.66 and 32.4, respectively, during transition from DP to PP (Table 2).

Principal-coordinate analysis (PCO) showed clear shifts in the active rumen bacterial and eukaryotic community structures during transition from LL to DP, followed by a re-shift of the PP microbiota closer to the LL over the transition period, as seen in the PCO plot with the first two principal-coordinates describing 46% (bacteria, Figure 1A) and 52.3% (eukaryotes, Figure 1B) of the total variations. These results were further confirmed with the analysis of similarities (ANOSIM) test, which revealed statistically significant differences in the active rumen bacterial (Global R = 0.683, *p* < 0.001) as well as eukaryotic (Global R = 0.339, *p* < 0.001) communities of different physiological stages.

### 3.2. Physiological Stage-Dependent Modifications in the Active Rumen Bacterial Communities

A total of 4974 unique OTUs specific to the V3–V4 region of bacterial 16S rRNA transcript amplicons were obtained. There were 673 unique bacterial OTUs found in LL samples, 1866 OTUs in the DP samples, and 1163 OTUs in PP samples. A total of 194 bacterial OTUs were shared between LL and DP, 212 OTUs between DP and PP, 442 OTUs between PP and LL, and 424 OTUs were “core bacterial OTUs” commonly found in the three physiological stages (Figure 2A). The active rumen bacterial community was comprised of 17 phyla, of which Proteobacteria (37.56%), Bacteroidetes (26.62%), Firmicutes (11.38%), Spirochaetes (5.90%), Fibrobacteres (2.02%), Verrucomicrobia (1.42%), and Tenericutes (1.25%) were the seven most abundant, with median relative abundance value indicated in brackets (Figure 2B, Appendix A).

The Fibrobacteres and Spirochaetes phyla showed similar trends and were more abundant in the PP (6.73–9.98%) as compared to the LL and DP (Figure 2B).

Proteobacteria was the most dominant bacterial phylum in our study, being more abundant in high-concentrate-fed (i.e., LL and PP) than high-forage-fed (DP) Italian Simmental cows. A similar trend was observed for the dominant Gammaproteobacteria class, Succinivibrionaceae family and the dominant unclassified genus of this family, showing a significant decrease in relative abundance from 26.93% in LL to 4.13% in the DP, followed by a significant increase to 17.93% during transition from DP to the PP (Appendix A). On the contrary, the two least abundant genera of the Succinivibrionaceae family, *Ruminobacter* (1.24–4.81%) and *Succinimonas* (0.01–1.06%), showed opposite trends, increasing (1.24 to 2.06%) or decreasing (0.03 to 0.01%), respectively, from LL to PP, but both were more abundant in the DP (Figure 3, Appendix A). Likewise, the least abundant Alphaproteobacteria class and its corresponding Rhodospirillales order also showed a significant increase from 0.26% in LL to 2.60% in the DP, followed by a significant decrease to 1.05% in the PP period (Appendix A).

Bacteroidetes was the second most abundant phylum in our study, showing no significant (*p* = 0.83) change in relative composition during different physiological stages as did the dominant Bacteroidia class and Bacteroidales order of this phylum (Appendix A). Nevertheless, significant differences were found at lower taxonomic levels, where the Prevotellaceae family (Figure 2B), and its representative *Prevotella* genus were relatively more abundant in LL and PP (~13.0–14.0%) as compared to the DP (~8.0%) (Figure 3). In contrast, unclassified Bacteroidales was more abundant in the DP (6.17%) as compared to the PP (2.77%) and LL (1.87%).

Firmicutes was the third most abundant phylum in our study. The relative composition of this phylum and its predominant lower taxonomic groups remained unaffected (*p* > 0.05) by dietary changes (Figure 2B, Appendix A). However, the second most abundant family within this phylum, Ruminococcaceae was more abundant in the DP (5.03%) as compared to the LL (2.81%) and PP (2.36%). Likewise, the three least abundant rumen bacterial phyla, namely Verrucomicrobia (0.96–4.30%), Elusimicrobia (0.25–2.84%), and SR1 (0.50–2.08%), and the lower taxonomic groups within these phyla were also more abundant in high-forage-fed DP cows than other physiological stages (Figure 2B, Appendix A).

The Fibrobacteres and Spirochaetes phyla showed similar trends and were more abundant in the PP (6.73–9.98%) as compared to the LL and DP (Figure 2B). Similar trends were observed for the dominant *Fibrobacter* and *Treponema* genera of these phyla (Figure 3 and Appendix A).

### 3.3. Physiological Stage-Dependent Modifications in the Active Rumen Eukaryotic Communities

The active eukaryotic community in the rumen was represented by a total of 256 unique OTUs specific to the 18S rRNA V9 gene region. There were 29 unique OTUs found in the LL samples, 101 OTUs in the DP, and 22 OTUs in the PP samples. Only 13 OTUs were shared between LL and DP; similarly, 13 OTUs between DP and PP, 14 OTUs between PP and LL, and 64 OTUs were “core eukaryotic OTUs” commonly found in the three physiological stages (Figure 4A). A total of 17 eukaryotic phyla was observed, of which the protozoal phylum Ciliophora was the most dominant, showing a median relative abundance value of 95.50% (Figure 4B); a similar result was reported previously [22]. At the phylum level, no significant differences were detected in the active eukaryotic community composition of high-concentrate-fed (i.e., LL and PP) and high-forage-fed (DP) Italian Simmental cows, except a slight decrease in the relative abundance of phylum Ciliophora during the transition from LL to the DP and PP (Figure 4B, Appendix A). However, significant differences were found at the family level, where the two most abundant families within the phylum Ciliophora, namely Ophryoscolecidae (61.85–82.55%) and Isotrichidae (13.73–31.00%), showed different patterns, with the former being more abundant in LL and PP, and the latter being highly abundant in the DP (Figure 4B).

At the genus level, *Entodinium* (*Ophryoscolecidae*) was the most dominant protozoal genus in all physiological stages, accounting for 76.26–82.30% of the total protozoal abundance in high-concentrate-fed PP and LL cows versus 58.43% in high-forage-fed DP cows (Figure 5). Notably, a high-forage diet in the DP resulted in a complete shift in the protozoal community at the genus level, as indicated by a high dominance of the protozoal genera *Dasytricha* (*Isotrichidae*), *Eudiplodinium*, and *Ostracodinium* (*Ophryoscolecidae*) in the DP compared to the other physiological stages (Figure 5). The other least abundant protozoal families within the phylum Ciliophora, such as Buetschliidae (0.43–0.64%), Cycloposthiidae (0.02–0.15%), and Spirodiniidae (0.00–0.06%), and their lower taxonomic groups, were not affected by dietary changes (Figure 4B, Appendix A).

The least abundant protozoal phylum Amoebozoa (0.52–1.35%) and other eukaryotic phyla Chytridiomycota (0.60–4.44%), Rhodophyta (0.81–1.34%), and Metamonada (0.36–1.08%) also remained unaffected by physiological stage-dependent dietary changes (Figure 4B, Appendix A). Nevertheless, three families within the phylum Amoebozoa, namely Actyosteliidae (0.02–0.73%), Dactylopodida (0.01–0.32%), and Entamoebidae (0.00–0.08%), showed significant changes in relative composition with dietary changes during different physiological stages (Figure 4B). Dactylopodida and its representative *Angulamoeba* genus were more abundant in the LL and PP and almost absent in the DP samples. On the contrary, Actyosteliidae, Entamoebidae and their representative *Acytostelium* and *Entamoeba* genera, respectively, showed similar trends and were more abundant in the DP than PP and LL. Similar minor increases in the relative abundances of the Hypotrichomonadidae family and the *Trichomitus* genus of phylum Metamonada were observed during DP (Appendix A).

## 4. Discussion

Physiological stages of dairy cows and diet amendments during the transition period strongly affect the active rumen bacterial and protozoal communities. The rumen microbiota has a direct effect on host nutrient utilization affecting the metabolism and health of dairy cows. This symbiosis is complex and must be maintained in equilibrium for a healthy transition period. Indeed, the effective management of dairy cows requires knowledge of the active microbiota modification in the different physiological stages of dairy cows, related to different diets.

The use of an RNA template in a 16S metabarcoding analysis provides the relative amounts of the living microbial taxa forming the active part of the rumen microbiota.

The highest diversity of active rumen bacteria during the DP stage reported in the present study is in agreement with other DNA/RNA-based studies [48,49] and is probably related to the high forage content in the diet. High ruminal bacterial diversity with high-forage diets and low bacterial diversity with high-concentrate diets have been already reported [50,51]. Similar negative impacts of a high-concentrate diet on bacterial species richness and diversity have been discussed by Zhang et al. [52] and Pinto et al. [53]. Likewise, a significantly higher eukaryotic taxa diversity was observed during the DP in our study, which is in line with other studies [54,55], which reported that the high-forage diet supports high protozoal species diversity compared to the grain-based diet.

The overall dominance of the Proteobacteria phylum in our study is in agreement with the results of other RNA-based studies. Kang et al. [30] provided the first evidence of the high abundance of Proteobacteria (28.70%) at the RNA level, indicating higher importance of this group in rumen metabolic activities than previously ascribed based on DNA-level studies. A significant increase in Proteobacteria relative abundance was observed with increasing dietary concentrate proportions [49] and in high-energy-fed beef steers [34,56] at the RNA level. In line with other RNA-based studies, we found that by increasing energy content and the dietary corn silage proportion from 20.3 (% of dry matter) in DP to 32.0 and 47.4 (% DM) in PP and LL periods, respectively, the relative composition of their Proteobacteria phylum as well as the Succinivibrionaceae family of this phylum was also significantly increased. An increase in Proteobacteria abundance related to high energy diet has been reported by several studies [3,57,58,59], while others [9,10,60] have reported high abundance of Succinivibrionaceae. The bacteria belonging to the Succinivibrionaceae family play an important role in rumen succinate production through hydrogen utilization, which enables them to compete with hydrogenotrophic methanogens for substrate [61,62,63]. Although Proteobacteria are not considered to be the dominant rumen bacterial phylum at the DNA level, the high dominance of this phylum at the RNA level and the specific role of Succinivibrionaceae in succinate production (a propionate precursor), indicates that probably it is the main contributor of ruminal propionate production under a high-grain diet [64].

Within the Bacteroidetes phylum, the high abundance of Prevotellaceae in the rumen of high-concentrate-fed cows (i.e., LL and PP) in our study is in accordance with other studies [56,65], where this family comprised 17.9–35.18% of the total rumen microbial community of high-energy-fed cows. *Prevotella* species are strictly anaerobic and may comprise up to 60% of the total rumen bacterial population in silage-fed animals [66]. Ruminal *Prevotella* species digest starch, simple sugars, and non-cellulosic polysaccharides to produce succinate [67]. In addition, some ruminal *Prevotella* taxa are also involved in the metabolism of pectin, hemicellulose [68], protein, and peptides [69]. The use of high amounts of concentrates (rapidly fermentable carbohydrates) in the diet of LL and PP and the reduction in eating time could explain the high relative abundance of *Prevotella* during those periods, as observed by Kljak et al. [70].

Considering the Firmicutes phylum, the high dominance of Ruminococcaceae in the rumen fluid samples of DP is in agreement with other studies that also showed an increase in abundance of Ruminococcaceae members with an increase in forage contents of the diet [49,58]. Ruminococcaceae taxa play an essential role in cellulose and hemicellulose degradation [71,72] and in the ruminal biohydrogenation pathways [73], which could explain their high abundance in high-forage-fed DP cows in our study. The Elusimicrobia phylum is usually unique to RNA-based studies, indicating high activity of this phylum in the rumen [34]. This phylum is underrepresented or totally absent in the DNA-based datasets [74], which could be due to their unsuccessful DNA isolation [34], as it has been reported previously that the choice of DNA-extraction methods might have an impact on the taxonomic outcomes of rumen microbial communities [59].

The Fibrobacteres and Spirochaetes phyla were in our study more abundant in the PP as compared to the LL and DP, and similar trends were observed for the dominant *Fibrobacter* and *Treponema* genera of these phyla. An opposite pattern for these genera was reported recently [65,75]; however, both these studies used DNA as a template for defining communities.

Regarding the eukaryotic community, the protozoan phylum Ciliophora was the most dominant. A similar result was reported previously [19]. A high abundance of the *Entodinium* genus in high-concentrate-fed (LL and PP) cows in our study is similar to other studies, where a high dominance of *Entodinium* in concentrate-fed Korean cows [76], malt meal-fed cattle [77], high-corn silage and grain-fed cows [55,78], and concentrate-fed Jersey cows [79] was reported. A study by Zhang et al. [52] showed a linear increase in the abundance of the *Entodinium* genus with the increase in the amount of concentrates in the diet. Similarly, many culture-based studies also reported an increase in the number of *Entodinium* genus members with the increase in dietary concentrate proportion in ruminants [53,54]. The entodiniomorphid protozoa are highly resistant to low ruminal pH [53], possess an ability to engulf starch [80] and amylase activity [81], and thus play an important role in the removal of free starch from high-grain-fed ruminants [80].

Our study reported a significant increase in the relative abundance of the *Dasytricha* genus from 1.46% and 2.21% (LL and PP, respectively) to 14.28% (DP) with decreasing dietary corn silage proportions; however, the relative abundance of the *Isotricha* genus remained unaffected (*p =* 0.59) by dietary changes (Figure 5, Appendix A). The high abundance of *Dasytricha* in high-forage-fed DP cows in our study was similar to other studies, where the abundance of *Dasytricha* sp. increased with increasing dietary forage proportion [82,83] as well as in sugarcane-fed cattle [84] and was negatively impacted by dietary corn silage proportions [78]. The increased proportion of holotrich ciliates (*Dasytricha ruminantium* and *Isotricha* species) in the rumen of high-forage cows in our study was probably due to their strong adherent ability towards solid feed particles and the rumen wall, thus preventing them from being washed away from the rumen [85]. Another possible explanation could be an increased amount of soluble sugar from the meadow hay in the DP ration. This would be supported by the fact that the *Dasytricha* genera is characterized by prominent cellobiosidase and glucosidase activity but minor fibrolytic activity [86].

However, we cannot confirm this hypothesis, since the focus of our research was on forage diet modification related to the physiological state of the cow during the transition period. Thus, we focused on fiber content and not soluble sugar in the given diet. The use of a high-forage diet during the DP also increased the relative abundance of *Eudiplodinium* and *Ostracodinium* genera compared to the other physiological stages (Figure 5, Appendix A). A linear decrease in the relative abundance of *Ostracodinium* with an increase in the amount of dietary concentrate proportion has been previously reported [52]. The species of *Eudiplodinium* and *Ostracodinium* possess high cellulolytic activity [54,87]. The preferential cellulose uptake by *Eudiplodinium maggii* and its ability to rapidly digest cellulose and synthesize amylopectin from cellulose [88] highlights the importance of this protozoal genera in cellulose metabolism of high-forage-fed dairy cows.

## 5. Conclusions

The approach of the RNA-based amplicon sequencing method allows for an accurate prediction of metabolically-active rumen bacterial and protozoal communities. Particularly, in the study on the effects of different factors affecting the ruminal microbial populations (i.e., diets, stress, climatic conditions, pathologies, etc.), the ability to distinguish between active and inactive, lysed or dead microorganisms is essential. The results of this experiment showed that the physiological stage and the relative diet modification affects significantly the composition of the bacterial and protozoal communities, especially during the dry period, when the largest changes of the diet composition are required.

## Figures and Tables

**Figure 1 microorganisms-09-00754-f001:**
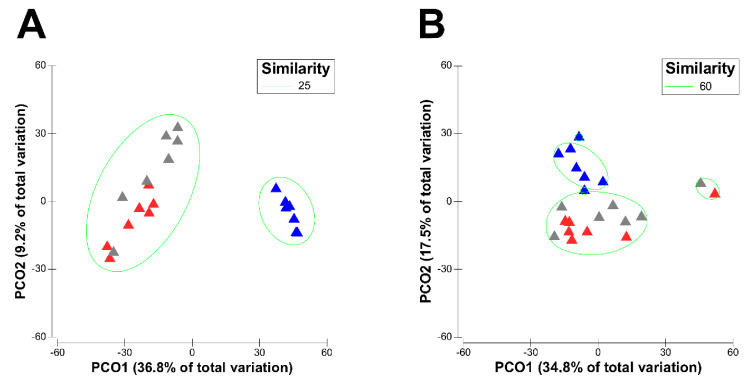
Principal-coordinate analysis (PCO) of the active rumen bacterial (**A**) and eukaryotic (**B**) communities during different physiological stages. The bacterial OTU abundance data were normalized by total sum normalization, while eukaryotic OTU abundance data were transformed by Hellinger transformation (total sum normalization combined with square root transformation) prior to the calculation of Bray–Curtis dissimilarity matrices. The three physiological stages are indicated by different colored triangles: red triangles, LL; blue triangles, DP; grey triangles, PP. The LL and PP is a major cluster with 25% similarity in the bacterial dataset, and with 60% similarity in the eukaryotic dataset.

**Figure 2 microorganisms-09-00754-f002:**
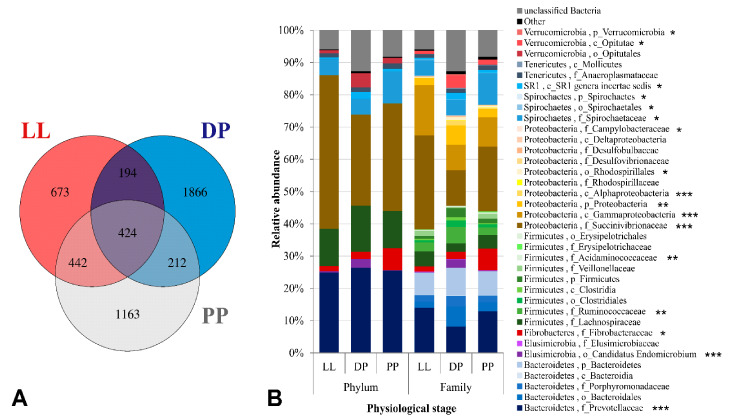
Distribution of bacterial-OTUs and active rumen bacterial taxa in different groups. (**A**) Venn diagram of OTUs specific to the V3–V4 region of bacterial 16S rRNA. The bacterial OTUs identified during different physiological stages were grouped as either unique or shared OTUs between LL, DP, and PP microbiome. (**B**) Relative abundance (percent reads out of total reads) of nine bacterial phyla and 36 bacterial families are shown. Phyla with relative abundance <1% were grouped as “Other”. Usually, family (f) was indicated; when family was not identified, lower identified taxa were indicated. The significantly different bacterial taxa between physiological stages are indicated by *p* values (ANOVA). * *p* ≤ 0.05, ** *p* ≤ 0.01, *** *p* ≤ 0.001.

**Figure 3 microorganisms-09-00754-f003:**
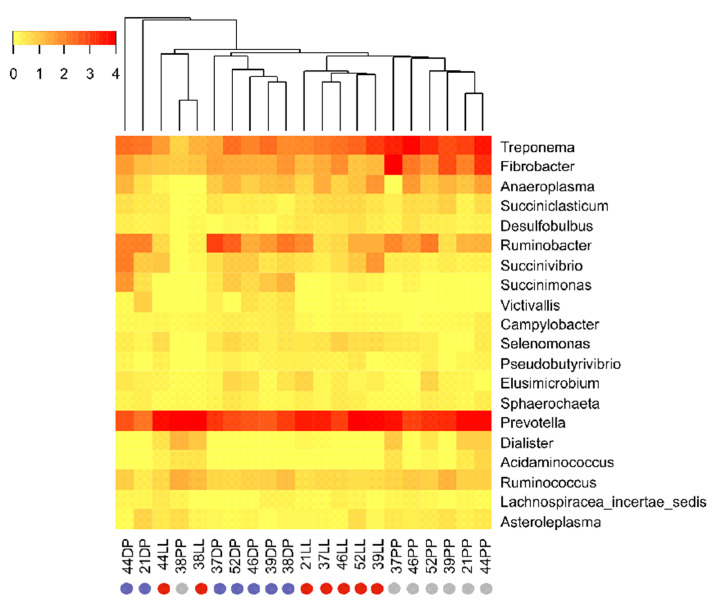
Heatmap at genus-level of active rumen bacterial communities of different physiological stages. The twenty most abundant bacterial genera (excluding the unclassified bacteria) during LL, DP, and PP were included in the heatmap.

**Figure 4 microorganisms-09-00754-f004:**
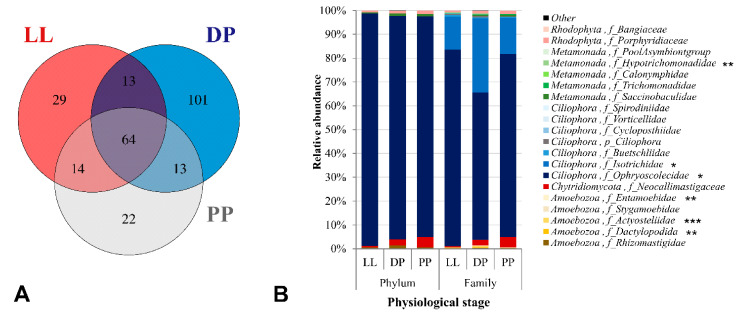
Distribution of eukaryotic-OTUs and active rumen eukaryotic taxa in different groups. (**A**) A total of 256 OTUs specific to the V9 region of eukaryotic 18S rRNA, identified during different physiological stages, were included in the Venn diagram and grouped as either unique or shared OTUs between the LL, DPs and PP microbiome. (**B**) Relative abundances of five eukaryotic phyla and 20 eukaryotic families are shown. Phyla with relative abundance <1% were grouped as “Other”. The significantly different bacterial taxa between physiological stages are indicated by *p* values (ANOVA). * *p* ≤ 0.05, ** *p* ≤ 0.01, *** *p* ≤ 0.001.

**Figure 5 microorganisms-09-00754-f005:**
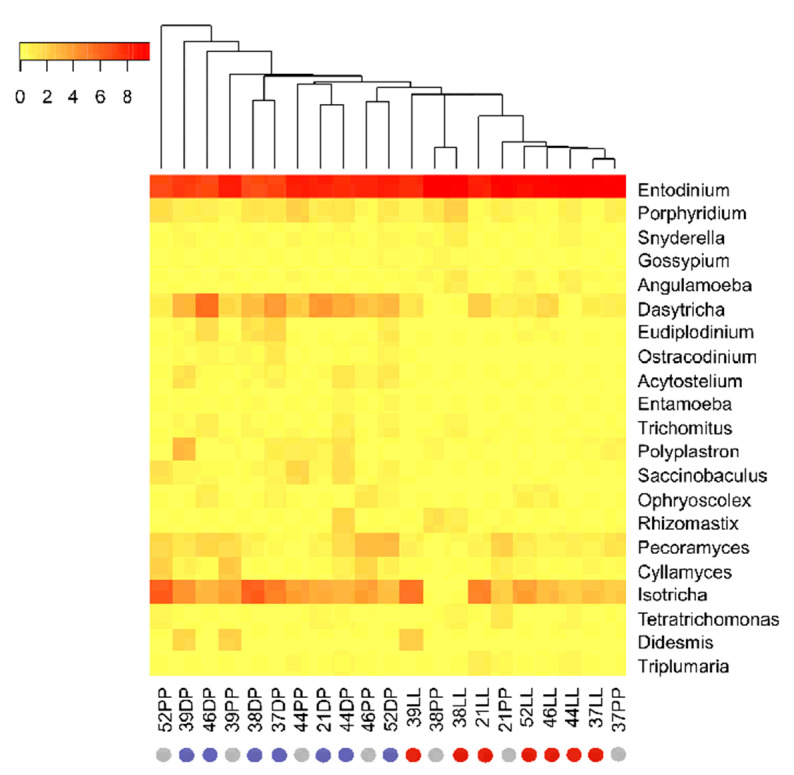
Heatmap at genus-level of active rumen eukaryotic communities of different physiological stages. The twenty most abundant genera (excluding the unclassified taxa) during LL, DP, and PP were included in the heatmap.

**Table 1 microorganisms-09-00754-t001:** Ingredients (% of dry matter) and composition (% of dry matter) of the total mixed rations (TMR) fed to the cows during three different physiological stages.

Item	Physiological Stage ^1^		
	LL	DP	PP
Ingredients			
Meadow hay	13.1	24.6	2.5
Alfalfa hay	17.4	-	20.6
Maize silage	47.4	20.3	32.0
Wheat straw	-	37.8	1.7
Dry sugar beet pulp	-	-	2.4
Protein mix ^2^	10.6	16.1	17.7
Energy mix ^3^	8.9	-	19.3
Extruded linseed	1.1	-	0.4
Fat supplement ^4^	-	-	1.2
Vit–min mix ^5,6^	1.4 ^5^	1.2 ^6^	2.2 ^5^
Chemical composition			
Dry matter	50.2	63.8	55.1
Crude protein	13.1	13.0	15.7
Lipids	3.4	3.2	4.2
Starch	21.1	7.1	24.0
NDF	39.4	57.0	33.6
ADF	23.4	36.0	21.1
MFU no./kg DM ^7^			

^1^ LL = late lactation; DP = dry period; PP = post-partum period; ^2^ protein mix: 50% of decorticated sunflower meal, 35% of toasted full-fat soybean meal, 12% of 44 soybean meal, 3% of corn meal; ^3^ energy mix: 66.67% of corn meal and 33.33% of barley meal; ^4^ fat supplement: Multifat (Nutristar, Reggio Emilia, Italy) contained linseed, calcium soaps obtained by palm oil, carob germ, corn (CP = 16.0%; lipids = 42.50%); ^5^ vitamin–mineral supplement for lactating cows: Milk H (Tecnozoo S.R.L., Torreselle di Piombino Dese, PD, Italy) contained per kilogram: 330,000 IU of vitamin A, 60,000 IU of vitamin D3, 2000 mg of vitamin E, 100 mg of vitamin B1, 75 mg of vitamin B2, 50 mg of vitamin B6, 0.3 mg of vitamin B12, 12,000 mg of niacin amide, 8 mg of biotin, 300 mg of Fe, 150 mg of I, 30 mg of Co, 300 mg of Cu, 2000 mg of Mn, 3000 mg of Zn, 15 mg of Se; ^6^ vitamin–mineral supplement for dry cows: Tecnofertil (Tecnozoo S.R.L., Torreselle di Piombino Dese, PD, Italy) contained per kilogram: 315,000 IU of vitamin A, 200,000 IU of vitamin D3, 1500 mg of vitamin E, 300 mg of vitamin K3, 50 mg of vitamin B1, 20 mg of vitamin B2, 50 mg of vitamin B6, 1 mg of vitamin B12, 4000 mg of niacin amide, 1500 mg of Fe, 200 mg of I, 30 mg of Co, 50 mg of Cu, 2000 mg of Mn, 6000 mg of Zn, 20 mg of Se; ^7^ Milk forage unit (MFU) no. per kg of dry matter.

**Table 2 microorganisms-09-00754-t002:** Comparison of various alpha-diversity estimators of active rumen bacterial and eukaryotic communities of different physiological stages.

Index ^1^	Bacteria					Eukaryotes				
	LL	DP	PP	SEM ^2^	*p*-Value ^3^	LL	DP	PP	SEM ^2^	*p*-Value ^3^
Richness										
Chao 1	464 ^b^	692 ^a^	543 ^a,b^	36.50	0.026	47.0 ^b^	68.0 ^a^	40.0 ^b^	3.47	<0.001
Evenness										
Shannon	0.71 ^b^	0.84 ^a^	0.74 ^b^	0.02	<0.001	0.39	0.48	0.39	0.03	0.3000
Simpson	0.93 ^b^	0.98 ^a^	0.96 ^a,b^	0.01	0.002	0.62	0.77	0.58	0.04	0.110
Diversity										
Shannon	4.35 ^b^	5.50 ^a^	4.66 ^b^	0.14	<0.001	1.45 ^a,b^	2.01 ^a^	1.43 ^b^	0.11	0.031
Inverse Simpson	16.4 ^b^	86.6 ^a^	32.4 ^a^	8.56	<0.001	3.18	4.66	3.02	0.33	0.081

^1^ Alpha-diversity estimators among amplicon-sequencing datasets; ^2^ standard error of the mean; ^3^
*p*-values were obtained doing one-way analysis of variance (ANOVA), and *p* < 0.05 shows significant effect of the physiological stage. For pairwise comparisons, Wilcoxon rank test was used. ^a,b^ Means within a row with different superscripts differ significantly (*p* < 0.05).

## Data Availability

The data presented in this study are available on request from the corresponding author.

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
