# Peer review of "Active Rumen Bacterial and Protozoal Communities Revealed by RNA-Based Amplicon Sequencing on Dairy Cows Fed Different Diets at Three Physiological Stages"

_microorganisms, 2021, doi:10.3390/microorganisms9040754_

Round 1

Reviewer 1 Report

Active rumen bacterial and protozoal communities revealed by RNA-based amplicon sequencing on dairy cows at different 3 physiological stages

This manuscript addresses an important topic, and it is generally well written with only a few errors. However, there are a number of major issues regarding the methodology and analysis, which are listed as follows:

Line 93: While esophageal tubing represents a less invasive method compared to ruminal canulations, with this approach we cannot obtain a representative sample of the ruminal contents. A good representative sample of ruminal contents is usually conducted by collecting and compositing samples from different locations within the rumen. Therefore, esophageal tubing can be seriously biased when obtaining a representative sample of ruminal contents. In particular, sampling of the populations attached to feed particles is not possible with his approach. Can authors discuss this issue? How the results could be biased by sampling methods and what would be the implications on the interpretation of results?

The results and discussion section seems simply a description of results. Authors briefly compared results with other reports regarding the microbial community structure due to diet differences. However, there is virtually no discussion regarding the association and implications of the findings with the physiologic stages of cows evaluated (Lines 87-99). Thus, authors need to discuss how the findings can be associated with physiologic stage. For example, can those changes in the microbial communities with related to milk production? Can those changes be associated with body tissue accretion or mobilization? Can those changes in the microbial communities be associated with energy balance in the animals (as they go from a positive to a negative energy balance from late gestation to lactation)? Otherwise, there is no new contribution of the results, as the effect of diet on the microbial communities has already been extensively reported.

Line 116-117: Please, report values of centrifugation in centrifugal force (g). To obtain a bacterial pellet from ruminal contents, high centrifugation speed has been commonly used, around 20,000 × g for 15 min (https://doi.org/10.4141/A04-054). In addition, ruminal protozoa isolation has been widely used and described (https://doi.org/10.3168/jds.S0022-0302(05)72885-X). Is there any reason why protozoa were not first isolated? If protozoa and bacteria are isolated separately first, this would allow extraction of DNA that would originate from the corresponding microbial community.

Line 215: Please, check accesion number. It could not be accesed at the moment of revision. Maybe the uploaded material has not been released yeat.

Data on microbial taxa presented in Figure 2 and 4 would be more illustrative if listed in Tables with corresponding P values (similar to what is presented in supplementary material). The way they are currently presented does not show the p-values, therefore statistical significance in figures cannot be determined. Please, list in tables with p-values and standard error of the mean (or any other source of variation) for a more accurate evaluation.

Reviewer 2 Report

In my opinion this work presents an original and very relevant approach to the characterization of rumen microbiotic communities, taking into account, not only the presence of the various taxa, but the activty state of the sampled organisms. It is well structured, its experimental design is appropriate, but it has 2 points that I consider that may be improved  before its acceptance:

  • (line 434-443 - Results and Discussion section) - In my opinion, this period should be improved. Although the sugar in the diets has not been analyzed (at least its content has not been presented), in my opinion the difference that the authors found for the genus Dasytricha may be mainly related  to the probable greater amount of sugar in the DP period diet, that came from the meadow hay, and not mainly  due to the difference in NDF content, also derived from the hay. Holotrich ciliates metabolize mainly non-structural carbohydrates and soluble sugars, which are immediately available in the environment, and do not have an active and strong fibrolytic activity such as, for example, genera such as Eudiplodinium and Epidinium, that are typically cellulolitics. Moreover, there are differences between the genera Dasytricha and Isotricha genera, the latter also being able to metabolize starch, which justifies the absence of differences between periods/diets for Isotricha and only diferences found for Dasytricha. Dasytricha also has cellobiosidase activity, but less than glucosidade, due to its the preference for soluble sugars. In my opinion, is this difference in carbohydrates metabolism with the preference by soluble sugars  by Dasytricha  that also mainly justifies the results of the articles that the authors present to support their discussion (one of them, 87, in which Dasytricha increased with the inclusion of sugar cane, illustrates this reasoning well).  Meadow hay often, but depending on the phenological phase of the plant in which it was harvested, can be quite rich in soluble sugars, which may more strongly justify the results obtained than the justification the authors present. A classic article on the metabolism of Holotrich Protozoa (Williams 1986 - Rumen Holotrich Ciliate Protozoa, Microbiological Reviews, March 1986) is attached to help the authors to improve this point of the discussion. Also more recent literature can be usefull as Newbold at al 2015 ( Front. Microbiol., 26 November 2015) or Firkins et al.,  (Front. Microbiol., 28 February 2020).
  • In my opinion the conclusions section may also be improved. The authors should pay attention and made it clear that the effects of the physiological stage of cows on rumen microbiota activity are mixed with the effects of the respective diet fed to the cows in each stage and are a consequence of those.  Hence the differences found cannot be assumed as to be only caused only by the physiological state of cows.  Please, pay attention to this point.

Nice work! Congratulations.

Reviewer 3 Report

The reviewed manuscript entitled “Active rumen bacterial and protozoal communities revealed by RNA-based amplicon sequencing on dairy cows at different physiological stages” reports ruminal bacteria and protozoa community’s characteristic during late lactation, dry period, and postpartum stages. Authors in reviewed MS collected rumen fluid for molecular analysis from seven Italian Simmental cows.

The general idea of the reviewed MS is with line of the Microorganisms journal scope and is interesting for the readers.

Although the presented topic is interesting and the MS shows scientific potential, I have some doubts that I listed below:

Please introduce hypothesis in Introduction part which can help to improve scientific value of MS. I also suggest to rewrite part from L 37 to 51. Information included in L37 to 51 is not enough informative. In my opinion it should be mainly underlined that only small percentage of all microorganisms inhabiting rumen has been well described. (For example: Pers-Kamczyc, et al., (2011). Journal of Animal and Feed Sciences, 3(20). DOI: https://doi.org/10.22358/jafs/66189/2016 informed that only up to 10%. Please find other references and add into this part). Therefore, advanced molecular methods like RNA-based amplicon sequencing can increase our understanding of processes occurring within the rumen. Above information will be very well combined with information from L 53 and can help to create proper hypothesis.

Did the Authors also perform a quantitative analysis of the microorganisms or only semi-quantitative one? Please clearly state. If quantitative analysis, please in discussion part (if it is possible to present quantitatively) introduce some information, for example what percentage of all rumen microorganisms after obtained results from described study are known. This information will be very helpful to better understand rumen metabolism regulated by microorganisms and will be very useful for other researchers.

The conclusion has to be focused on obtained results and only main message has to be present. If I properly understand no statistical comparison has been made, because of it please rewrite this part of MS and avoid expression: “compared to DNA-based methods”. Conclusion should be related to added hypothesis.

Please clearly state how many rumen samples were analysed? I hope that more than one from each cow in each period.

Please remember that in the future study only two, not four, layers of cheesecloth have to be used.

Considering my concerns listed above, I do recommend the evaluated MS for publication in Microorganisms but only after major revision.

Round 2

Reviewer 1 Report

Line 121. Please, report value of centrifugation speed as “speed” × g.

Line 128 and elsewhere. Same comments as above. In addition, please be consistent with the use of “minutes” vs “min”. Either spell the word or abbreviate it.

Line 133 and elsewhere. Centrifugation speed in g force.

Author Response

All changes suggested by the Reviewer 1 have been accepted. 

Reviewer 3 Report

In my opinion, the MS was improved and answers for my doubts were addressed.

I recommend the evaluated manuscript for publication in Microorganisms.

Author Response

Thank you very much.